# Investigations into the Photocatalytic and Antibacterial Activity of the Nitrogen-Annealed Titanium Oxide/Silver Structure

**Jun-Kai Zhang, Kui-Shou You, Chen-Hao Huang, Pin-Jyun Shih and Day-Shan Liu ***

Institute of Electro-Optical and Material Science, National Formosa University, Yunlin 632301, Taiwan
* Correspondence: dsliu@sunws.nfu.edu.tw; Tel.: +886-5-6315665

**Abstract:** In this study, a thin silver (Ag) layer was evaporated onto the anatase-titanium oxide (TiO$_x$) film. This structure was then annealed at various temperatures under nitrogen ambient to realize the Ag nanoparticles formed on the TiO$_x$ surface. The photocatalytic activities of these TiO$_x$/Ag structures to decompose pollutants were determined from the rate constant while they were applied to decolorize the methylene blue (MB) solution in the presence of the UV light irradiation. According to the investigations on their surface bond configurations, the Ag nanoparticles were favorable for the transformation of the Ti$^{4+}$ into the Ti$^{3+}$ state in the TiO$_x$ film, which functioned to prohibit the recombination of the photogenerated electron-hole-pairs on the TiO$_x$ surface. The exposed TiO$_x$ surface distributed over the 500 °C-annealed TiO$_x$/Ag structure performed an increase of about 40% in the rate constant compared to the individual TiO$_x$ film. Moreover, this surface morphology composed of the anatase-TiO$_x$ structures and Ag nanoparticles which was abundant in the oxide-related radical and Ag$^+$ chemical state also showed a perfect antibacterial efficiency against *Escherichia coli*.

**Keywords:** anatase-titanium oxide film; silver nanoparticles; photocatalytic activity; TiO$_x$/Ag structure; antibacterial efficiency

## 1. Introduction

In the past two decades, environmental pollution and global warming issues have garnered more attention. Regarding environmental pollution issues, the developments and research on the materials that decompose or reduce toxic gases and organic solvents emitted from industries have become a critical topic. Various technologies such as adsorption, electrocoagulation, ion exchange, and photocatalysis, have been applied for the removal of pollutants [1–4]. Among them, photocatalysis is the process that changes the rate of a chemical reaction by introducing electron-hole pairs (ehps) on the material's surface while it is exposed to light. The photogenerated ehps provides complex photooxidants such as hydroxyl radical (·OH), superoxide radical (·O$_2^-$), and hydrogen peroxide (H$_2$O$_2$) that exhibit strong oxidation for decomposing environmental pollutants [5]. The advantages of the photocatalysis over other technologies are such that the photogenerated ehps induced by the light irradiation on the photocatalyst are highly reliable and reproducible. Accordingly, investigations into photocatalytic materials as well as to strengthen its activity under light irradiation become one of the main projects of the current research. Regarding materials suitable for photocatalysts, titanium oxide (TiO$_x$) is the most popular candidate among the photocatalytic materials for being capable to decompose both the organic and inorganic pollutants due to its chemical stability, optical transparency at visible wavelength, high refractive index, non-toxic nature, and low cost. In general, the photocatalytic activity of the TiO$_x$ material is typically affected by three factors: the crystalline structure, the specific surface area, and the recombination rate of the photo-generated ehps. In terms of the crystallinity, the formation of the anatase-TiO$_x$ phase is crucial for exhibiting a high degree of the photocatalytic activity [6–8]. For the specific surface area, the processes to

construct a nanotextured surface or to fabricate nanospheric particles to enrich the reaction area exposed to the environmental pollutants are continually developed to improve the photocatalyticity of the $TiO_x$ [9–12]. In addition to achieving the anatase crystallinity and enriching the specific surface area of the $TiO_x$ film, the researchers also focused on separating or avoiding the recombination of the photo-generated ehps. Basically, there are two ways to prolong the existence of the photo-generated ehps in the $TiO_x$ film: engineer the heterojunction structure with energy band discontinuity using metals or metal oxides, such as Ag, NiO, $Cu_2O$, $SnO_2$, $WO_3$, or ZnO, contact to the $TiO_x$ film [13–17] or build up a defect state in the $TiO_x$ energy bandgap by introducing extrinsic impurities, such as C, N, Pd, or F dopants, which may also be beneficial for allowing the onset of the photocatalyticity under visible wavelength irradiation [18–21]. In addition to enhancing the film's photocatalyticity for purifying environmental pollutants, researchers also are interested in activating extra functions of the contact materials or dopants in the $TiO_x$ film to strengthen the resulting products in saving the troublesome problems of daily life. Considering the demands of daily life, efforts to ensure environmental sanitation and food safety are critical. A number of organic and inorganic materials have been employed as effective antimicrobial and/or antifungal agents to avoid the growth of viruses and bacteria [22–27]. They allow the excessive formation of reactive oxygen species, leading to oxidative stress and subsequently damaging or altering the cell structure and function. Among them, Ag is the most widely used material to fill this role not only because of its high biocidal activity against a broad range of microorganisms and viruses, but also because it can be simply nanosized via various processes to optimize the reaction efficiency [28–30]. To summarize the discussions mentioned thus far, $TiO_x$ and Ag materials seem to be the best candidates for simultaneously resisting environmental pollution and infection. As a result, a $TiO_x$/Ag composite system with both quality photocatalytic and antibacterial activities had been announced. For instance, Sheikh et al. fabricated the $TiO_2$ nanofibers containing Ag nanoparticles using an electrospinning process. They found that the Ag nanoparticles with diameters of 5 to 10 nm were dispersed well in/on the $TiO_2$ nanofibers after calcining at a temperature of 600 °C in air ambient for one hour. The resulting inorganic $TiO_2$/Ag nanofibers showed good stability for biomedical applications [31]. Song et al. synthesized a $TiO_x$ containing Ag nanoparticles onto the commercially Ti substrate using a reactive magnetron cosputtering system. The shape and size of the Ag nanoparticles dispersed in the $TiO_x$ film affected by the power applied to the Ag target as well as the bias imposed on the substrate were investigated. The results indicated that the substrate bias caused an increase in the small-sized Ag nanoparticles while the metallic Ag content decreased and oxidized Ag increased as the power applied to the Ag target increased. The reduction in the size of the Ag nanoparticles and the fast ion release from the metallic Ag was responsible for optimizing the structure's antibacterial activity [32]. Bian et al. synthesized the $Ag/ZnO/AgO/TiO_2$ composite by chemical precipitation and hydrothermal method. They found that the existence of the AgO and Ag materials sandwiched by the ZnO and $TiO_2$ nanoparticles was facilitated for trapping and transferring the electron carriers, and thereby lengthening a longer lifetime of the photo-generated ehps. The resulting composite performed a kinetic rate constant doubly higher than that of the $TiO_2$ nanoparticles as it was employed to photocompose the rhodamine B (RhB) solution (R2-1, [33]). Deekshitha et al. addressed a novel liquid-based method to synthesize $TiO_2$ embedded $AgO/Ag_2O$ nanocomposite. They demonstrated that the nanocomposite was activable both by UV and visible light irradiation due to the presence of the AgO and $Ag_2O$ functioned to reduce the bandgap of the $TiO_2$ nanoparticles. The $AgO/Ag_2O@TiO_2$ nanocomposite performed highly crystalline with a bandgap energy of 1.75 eV without any calcination and almost completely degraded 100 ppm Reactive Blue (RB) 220 dye under visible lamp irradiation for 90 min (R2-3, [34]). Kacprzyńska-Gołacka et al. presented a $TiO_2$ + AgO coating using Ti and Ag targets through the reactive magnetron cosputtering technology. They discussed the size of the Ag/AgO nanoparticles and the percentage of the elemental composition of the $TiO_2$ + AgO coating, both of which were affected by the applied power on the targets.

The results demonstrated that the $TiO_2$ + AgO coating showed complete inhibition of the growth of *Escherichia coli* (*E. coli*) and *Bacillus subtilis*. Moreover, this coating structure also performed good photocatalytic activity when it was irradiated under both ultraviolet (UV) and visible lights [35]. Thukkaram et al. prepared an Ag nanoparticles-loaded $TiO_2$ coating on a Ti disc using plasma electrolytic oxidation. They studied the bactericidal efficiency of the oxidized $TiO_2$ coating affected by the concentration of the Ag nanoparticles in the electrolyte. The resulting coatings showed that the bactericidal effect could be enhanced by increasing the amounts of the Ag nanoparticles, which was helpful for releasing more $Ag^+$ ions into the bacterial solution [36].

Though the $TiO_x$/Ag system had been prepared by various technologies, studies on the $TiO_x$/Ag structures under thermally annealing process are rare reports. The annealing process affected on the photocatalytic and antibacterial activity of the $TiO_x$/Ag system as well as its thermal stability are both critical. Accordingly, in addition to the research to modify the photocatalytic surface of an anatase-$TiO_x$ film with quality antibacterial features at the same time, an investigation on the structural evolution of the $TiO_x$/Ag structure influence by the annealing process also was carried out. For this purpose, a thin Ag layer as the antibacterial agent was evaporated onto the anatase-$TiO_x$ film. To study the thermal process affected on the surface property, a post-annealing treatment under nitrogen ambient was then processed on the $TiO_x$/Ag structure and was expected to achieve the $TiO_x$ surface decorated with Ag nanoparticles. Surface morphologies, chemical bond configurations, and crystalline structures of the as-deposited and annealed $TiO_x$/Ag structures were investigated. The corresponding photocatalytic and antibacterial activities were examined as the samples were used to decompose the methylene blue (MB) solution and sterilize *E. coli* activated by a UV lamp irradiation.

## 2. Material Preparation and Experimental Procedure

Hydro-oxygenated amorphous $TiO_x$ films with a thickness of 200 nm were deposited onto silicon and quartz substrates (~10 × 10 $mm^2$) using the home-made plasma-enhanced chemical vapor deposition (PECVD) at a temperature of 200 °C. Titanium precursor vaporized from the titanium tetraisopropoxide [$Ti(OC_3H_7)_4$, TTIP] liquid was intermixed with pure oxygen gas in a mixbox and subsequently inlet to the deposition chamber. The deposition pressure, rf power, and gas flow rate of $TTIP/O_2$ gas mixture were controlled at 40 Pa, 100 W, and 120/20 sccm, respectively. The amorphous $TiO_x$ films were then annealed at 600 °C for 30 min under ambient oxygen to result in the film crystallization with the anatase phase that possessed the optimal photocatalytic activity. A detailed setup for the PECVD system and evidence of the film crystallization have been described and reported elsewhere [37]. Following this, metallic Ag films with a thickness of 20 nm were thermally evaporated onto the crystalline $TiO_x$ films surface. These $TiO_x$/Ag structures were then rapidly annealed at 300 °C, 400 °C, 500 °C, and 600 °C, respectively, under nitrogen ambient for 1 min. This annealing process was treated for realizing the formation of the Ag nanoparticles which was expected to enrich the heterojunction contact and enlarge the exposed surface of the $TiO_x$ film and Ag layer. Another set of 20 nm-thick Ag films directly deposited onto the silicon and quartz were also prepared to study the evolution of the Au layers processed by the same RTA treatment.

The film thickness of the $TiO_x$ film was measured using a surface profile system (Dektak 6M, Veeco, New York, NY, USA). Surface roughness and morphologies of the individual $TiO_x$ and Ag films as well as the $TiO_x$/Ag structures were examined using an atomic force microscopy (AFM, DI-3100, Veeco, New York, NY, USA) and a field emission scanning electron microscope (FE-SEM, JSM-6700F, JEOL, Tokyo, Japan) operated at 3 kV. The corresponded optical transmittance of these samples was examined by a UV-Vis-NIR spectrophotometer (UVD 3500, Labomed, Inc., Los Angeles, CA, USA). The associated crystalline structures were conducted using a grazing incident x-ray diffractometer (GIXRD) at 30 kV and 30 mA using Cu $K_{\alpha 1}$ radiation (D-500, Siemens, Munich, Germany). X-ray photoelectron spectroscope (XPS) with monochromatic Al $K_\alpha$ radiation (PHI Quantera

SXM$^{TM}$, ULVAC-PHI, Kanagawa, Japan) were employed to examine the chemical bond configurations distributed over the structural surface. The energy scale of the spectrometer was calibrated using the core level of C 1s at 284.6 eV. The photocatalytic activities of the TiO$_x$ films incorporated with the thin Ag layers as well as the individual TiO$_x$ and Ag films were evaluated as they were applied to decolorize an aqueous MB solution under an UV lamp irradiation. The initial concentration of the MB solution was controlled at 20 mgL$^{-1}$ with a pH value approximately of $7.0 \pm 0.1$. The photocatalysis on the MB solution was determined from the degradation of the solution concentration that were linked to the absorbance evolution at 665 nm measured using the UV-Vis-NIR spectrophotometer. The antibacterial activities were assessed with a plate-counting method used against *E. coli*. Microbiological tests were carried out as the samples were immersed into a nutrient broth, and the initial concentration of the *E. coli* bacteria was adjusted to $1.0 \times 10^6$ colony-forming unit (CFU)/mL by dilution. These specimens were then sterilized under the UV-lamp irradiation for 1 h. Subsequently, 0.1 mL of each dilution was taken and spread on the nutrient agar and then incubated at 37 °C for 24 h. The number of the bacterial colonies grown on the plates was counted (in CFU) and photographed. The UV-lamp (TLD 10 W/08, Philips, London, UK) employed to activate the specimens' photocatalyticity for decomposing the MB solution and to suppress the *E. coli* growth emitted the dominated wavelength of 365 nm (light ranging from 350 to 400 nm) with the power density controlled at 5 mW/cm$^2$.

## 3. Results and Discussions

Figure 1a–e show the surface roughness of the 20-nm thick Ag film directly deposited onto the silicon substrate and the films annealed at 300, 400, 500, and 600 °C under nitrogen ambient for 1 min, respectively, examined by AFM measurement. The as-deposited Ag film presented flat surface morphology with a root-mean-square surface roughness, R$_q$, of about 2.7 nm (Figure 1a). In contrast, the annealed films exhibited rough characteristics with a surface roughness greater than 10 nm. Moreover, the surface roughness decreased slightly to about 9.4 nm as the annealed temperature reached 600 °C (Figure 1e). The surface morphologies conducted from the FESEM observation for the Ag thin film before and after the thermal annealed treatment are given in Figure 2a–e. As shown in Figure 2a, the surface of the as-deposited Ag film exhibited island structures with closed wedge-like particles. For the surface morphologies of the annealed samples shown in Figure 2b–e, these wedge-like islands evolved into separated and round-like nanoparticles. The size of these Ag nanoparticles were abnormally distributed over the annealed surface. Moreover, the mean diameters of these nanoparticles calculated by the software ImageJ analysis were gradually decreased as the annealed temperature increased. A mean diameter of about $53 \pm 14.5$ nm was calculated as the Ag film annealed at 600 °C, while that of the nanoparticles annealed at 500 °C was $69 \pm 16.3$ nm. Incorporating the measurements of the surface roughness shown in Figure 1, the closed wedge-like islands observed from the as-deposited Ag film surface corresponded to a relatively smooth surface, while the separated Ag nanoparticles appearing on the annealed surface resulted in a marked increase in surface roughness. The optical transmittance and absorbance spectra of the as-deposited and annealed Ag thin films directly deposited onto the sapphire substrate are illustrated in Figure 3a and b, respectively. As can be seen from the optical transmittance of the as-deposited sample in Figure 3a, the opaque nature of the Ag film showed an optical transmittance lower than 40% around the visible wavelength (400~700 nm). For the samples annealed at 300, 400, and 500 °C, the transparency at the long and short wavelengths of the visible wavelength region was greatly improved, with a wide drop in the transmittance at around 450–600 nm. As the Ag film annealed at 600 °C, the optical transmittance around the visible wavelength was further improved to about 90% except for the drop of the optical transmittance to about 71% at the wavelength of 453 nm. Combined with the surface morphologies of the annealed samples shown in Figure 2b–e, the light leakage through the Ag nanoparticles boundaries was responsible for the optical transparency around

the visible wavelengths, while the transmittance drop at the visible wavelength could be ascribed to the localized surface plasmon (LSP) resonance emerging from the distributed Ag nanoparticles [38–40]. The position of the absorbance peaks shown in Figure 3b could be connected to the general size of the Ag nanoparticle distributed over the annealed samples, as quoted from the reports [41–43]. The blue-shift of the absorbance peak as the annealed temperature increased was linked to the general size of the Ag nanoparticles being decreased as the annealing temperature increased. Moreover, the decrease in the bandwidth of the absorbance peak also implied the improved homogeneity of the Ag nanoparticles [44,45]. As a consequence, the 600 °C-annealed sample that exhibited the narrowest absorbance bandwidth with the peak at 453 nm corresponded to the smallest and best homogenous nanoparticle size, as observed from Figure 2e.

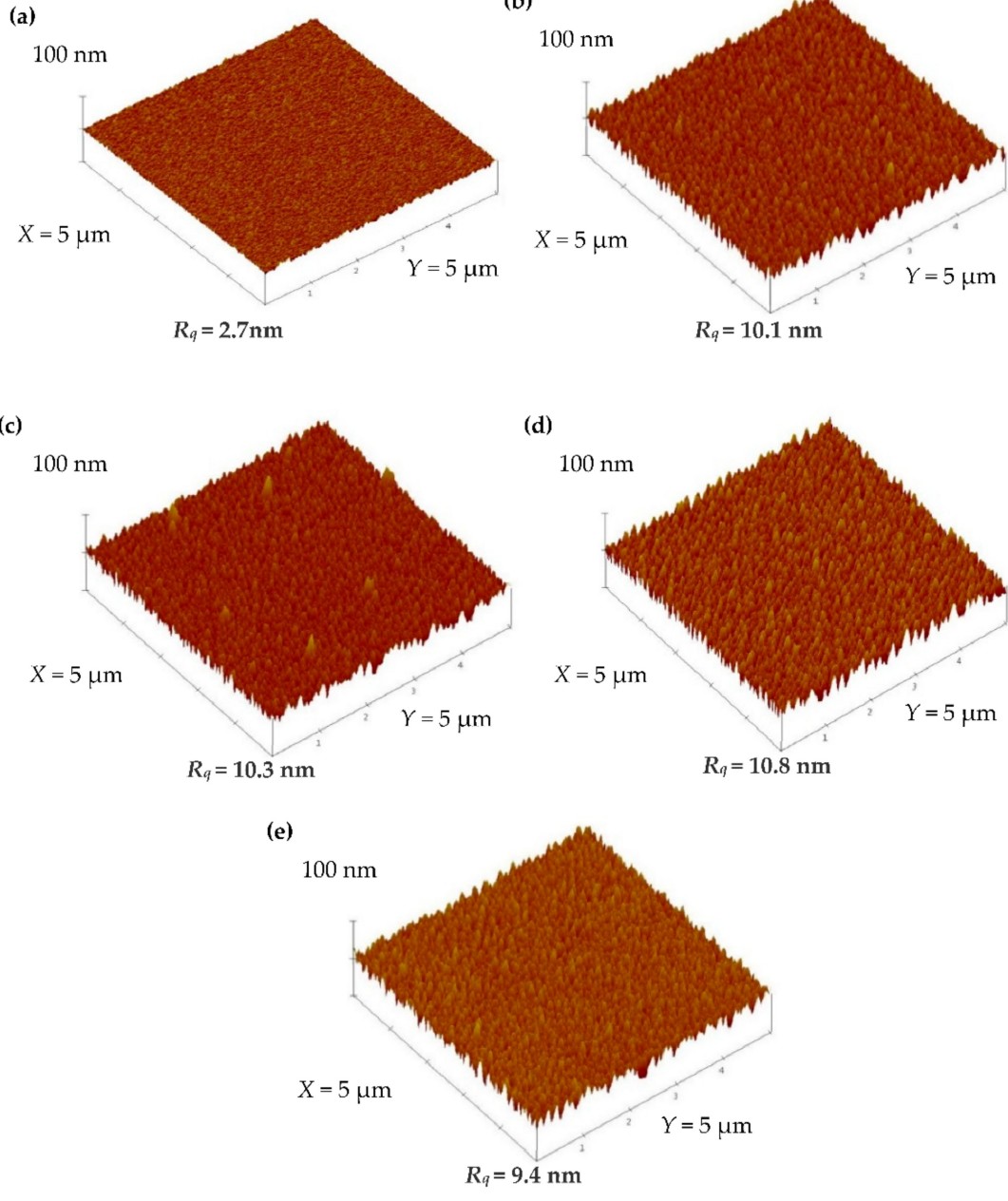

**Figure 1.** Surface roughness of (**a**) the as-deposited Ag film directly deposited onto the silicon substrate and the films annealed at (**b**) 300, (**c**) 400, (**d**) 500, and (**e**) 600 °C under nitrogen ambient for 1 min, examined by AFM measurement.

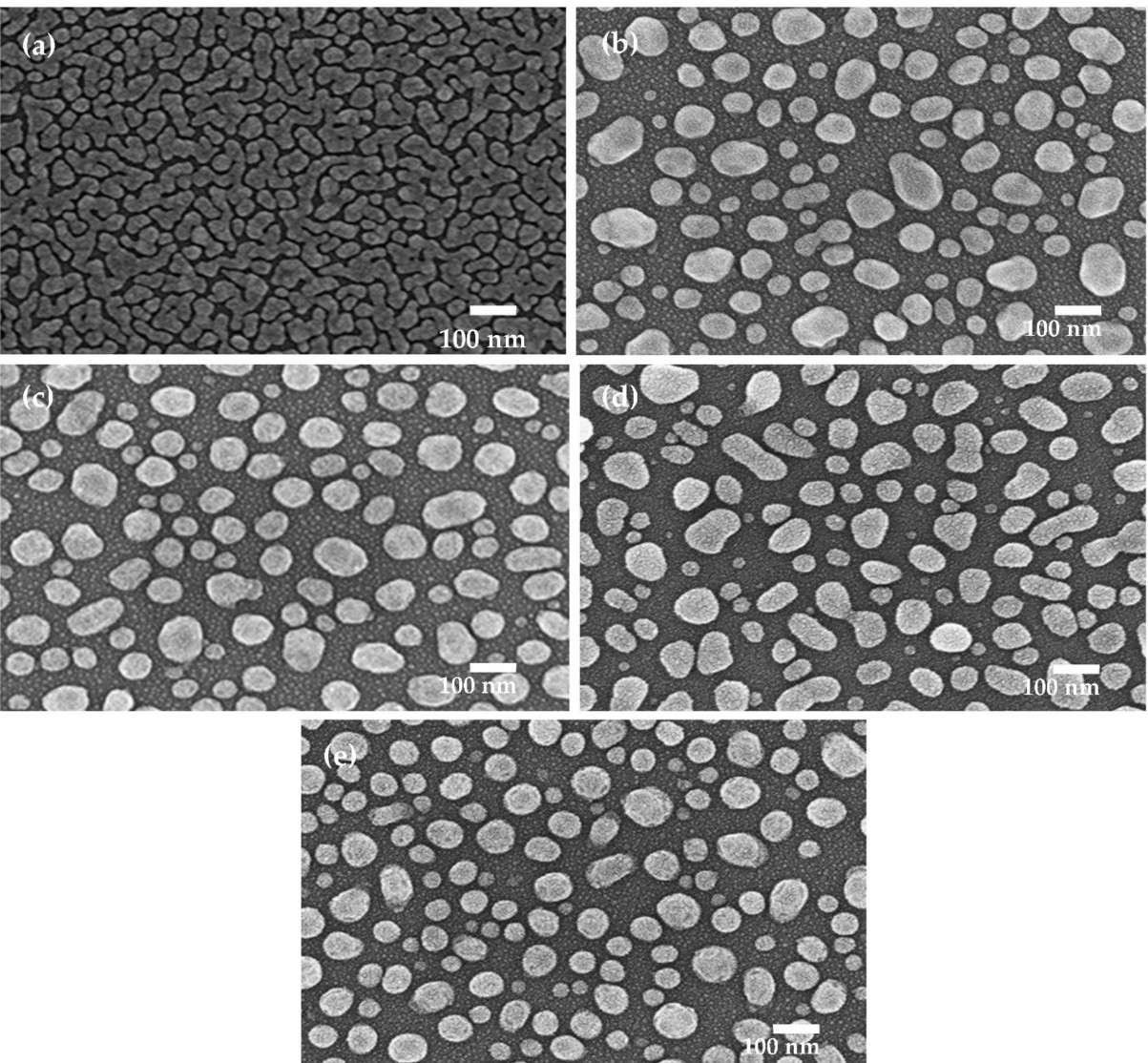

**Figure 2.** Surface morphologies of (**a**) the Ag film directly deposited onto the silicon substrate and the films annealed at (**b**) 300, (**c**) 400, (**d**) 500, and (**e**) 600 °C under nitrogen ambient for 1 min, conducted by FE-SEM observation.

According to these investigations, Ag nanoparticles were obtained from a thin Ag film treated by an adequate annealing process under nitrogen ambient. The specific surface area and optical transmittance around the visible wavelength of these Ag nanoparticles could be modified by controlling the annealed temperatures. This Ag layer with distinctive surface morphology and structure was then employed to alter the chemical and physical properties of the underlayer anatase-$TiO_x$ film. Figure 4b–f show the surface roughness of the as-deposited $TiO_x$/Ag structure and the structures annealed at 300, 400, 500, and 600 °C under nitrogen ambient for 1 min, respectively (the surface roughness of the individual $TiO_x$ film also is shown in Figure 4a for comparison). The PECVD-prepared $TiO_x$ film, after annealing at 600 °C for 30 min under oxide ambient, possessed a surface roughness of about 4.2 nm. As an Ag layer was deposited onto the $TiO_x$ film, the surface roughness was markedly increased to about 10.9 nm. A rougher surface that corresponded to an $R_q$ value of about 18.0 nm was measured from the $TiO_x$/Ag structure annealed at 300 °C for 1 min under nitrogen ambient. As the annealed temperature reached 400 °C, the resulting $TiO_x$/Ag structure presented the roughest surface with an $R_q$ value of about 18.5 nm (Figure 4d). The surface roughness of the $TiO_x$/Ag structure was then gradually decreased

to 16.2 and 15.5 nm while the annealed temperature was increased to 500 and 600 °C, respectively. The surface morphologies of the samples shown in Figure 4 are provided in Figure 5a–f. The grains distributed over the surface of the $TiO_x$ film shown in Figure 5a were closely packed with visible boundaries. Interestingly, the Ag film deposited onto the $TiO_x$ film (Figure 5b) was much different from the film directly deposited onto the silicon surface shown in Figure 2a. Nanoparticles and wedge-like islands were randomly and separately distributed over the $TiO_x$ surface. This implied that the surface property of the $TiO_x$ film facilitated the agglomeration of the Ag atoms to form islands and nanoparticles at a relatively low temperature. The appearance of the Ag islands and nanoparticles on the $TiO_x$ film surface therefore resulted in a marked increase in the surface roughness (Figure 4a) compared to the Ag layer directed deposited onto the Si substrate. Unlike in the nanoparticles distributed over the 300 °C-annealed Ag film while being directly deposited onto the silicon substrate shown in Figure 2b, the density of the Ag nanoparticles distributed over the $TiO_x$ surface under the same annealed temperature was obviously depressed, which might diffuse into the $TiO_x$ film through the grain boundaries. A large area of the under-layered $TiO_x$ film was thus observed from the surface morphologies of the 300 °C-annealed $TiO_x$/Ag structure (Figure 5c). The surface morphology combined with the scattered Ag nanoparticles and uneven $TiO_x$ grains resulted in a rougher surface roughness (~18.0 nm) compared to the 300 °C-annealed Ag film on the silicon substrate (~10.1 nm). The density of the Ag nanoparticles distributed over the $TiO_x$ surface decreased as the annealed temperature on the $TiO_x$/Ag structure increased. Figure 6 depicts the optical transmittance of the individual $TiO_x$ film, the as-deposited $TiO_x$/Ag structure, and the structures annealed at 300, 400, 500, and 600 °C. The individual $TiO_x$ film possessed an average transmittance higher than 90% around the visible wavelength, and then a sharp reduction in the near ultraviolet wavelengths was measured due to the absorption by the energy bandgap of the anatase-$TiO_x$ structure. For an Ag layer deposited onto the $TiO_x$ film, the resulting average optical transmittance markedly decreased to about 43% due to the injecting photons being obstructed by the Ag particles. The average optical transmittance was gradually increased as the annealed temperature on the $TiO_x$/Ag structure increased. An average optical transmittance of about 68% was measured from the structure annealed at the temperature of 600 °C. Considering the surface morphologies shown in Figure 5c–f, the increase in the transparency around visible wavelengths of these annealed $TiO_x$/Ag structures was ascribed to the decrease in the coverage of the Ag nanoparticles. Moreover, the size of the Ag nanoparticles distributed over the $TiO_x$ film surface that caused the LSP phenomena was identified as one of the reasons for the depressions in the optical transmittance over the visible wavelengths. The crystalline phases of the as-deposited and annealed $TiO_x$/Ag structures as well as the individual $TiO_x$ and Ag films conducted by the x-ray diffractometer measurement are illustrated in Figure 7. Two peaks identified as the anatase structure (denoted as "A" with the diffraction planes given in parentheses) according to JCPDS no.21-1272 could be observed from the XRD spectrum of the $TiO_x$ film [46]. The crystalline phase of A(101) was the preferred growth orientation in this anatase $TiO_x$ film. For the thin Ag layer, only a weak signal denoted as the Ag (111) that matched with the face centered cubic (fcc) structure according to the JCPDS no.04-0783 could be observed [47]. As this thin Ag layer was deposited onto the $TiO_x$ film, no significant peaks other than these peaks belonging to the anatase-$TiO_x$ and Ag phases appeared in the XRD spectrum. The XRD spectra of these annealed $TiO_x$/Ag structures also showed similar diffraction peaks to those of the as-deposited structure, indicating that little chemical reaction occurred between the $TiO_x$ and Ag materials during the Ag layer deposition and the subsequent annealed treatment. In addition, there was no evidence of the Ag atoms doped into the $TiO_x$ matrix as the peak position of the A(101) phase in the XRD spectra of the $TiO_x$/Ag structures was almost the same as the individual $TiO_x$ film. Regarding crystallinity of the Ag layer while deposited onto the $TiO_x$ film, the crystal size was smaller than that of the layer directly deposited onto the silicon substrate as derived from the full width of the half maximum (FWHM) of the Ag(111) peak according

to the Scherrer formula. Though the metal state was the dominated composition in the Ag nanoparticles, it still could not be ruled out the possibility of the oxidized state coexisted in the nanoparticles since the $Ag_2O$ signal might be overlapped by the Ag (111) peak. As a consequence, XPS analysis was carried out to investigate the compositions of the Ti, O, and Ag elements. The binding energy spectra related to the Ag 3d, Ti 2p, and O 1s core levels, as illustrated in Figure 8a–c, respectively, are provided to further elucidate the evolution of the chemical bond configurations on the surface of the as-deposited and 500 °C-annealed $TiO_x$/Ag structures (the individual Ag and $TiO_x$ films are also given as comparisons). Two distinctive peaks that were denoted as Ag $3d_{5/2}$ and Ag $3d_{3/2}$, with approximately 6 eV difference in the binding energy, could be seen in Figure 8a. The peak of the Ag $3d_{5/2}$ binding energy emerged from both the Ag film deposited onto the silicon and $TiO_x$ surface both was approximately 368.2 eV, whereas the $TiO_x$/Ag structure annealed at 500 °C caused a shift in the binding energy at 367.8 eV. According to previous literatures, the shift in the binding energy observed from the surface of the annealed $TiO_x$/Ag structure was ascribed to the evolution on the chemical stat from the metallic $Ag^0$ signal to the oxidized $Ag^+$ state [48–51]. In the Ti 2p spectrum (Figure 8b), two peaks assigned as Ti $2p_{3/2}$ and Ti $2p_{1/2}$, with a binding energy difference of 5.6 eV, were also observed. Compared to the individual $TiO_x$ film, the Ti $2p_{3/2}$ binding energy shifted from 458.9 eV to 458.6 eV as the $TiO_x$ film was covered by a thin Ag layer. This peak further shifted to 458.2 eV while the $TiO_x$/Ag structure annealed at 500 °C. The shift of the Ti $2p_{3/2}$ single toward a lower binding energy could be related to the increased formation of the titanium atoms from the $Ti^{4+}$ to the reduced $Ti^{3+}$ state upon the Ag layered deposition and the structural annealed treatment, as referred to the reports [52–56]. For the O 1s spectrum shown in Figure 8c, the surface of the individual $TiO_x$ film emerged a somewhat asymmetrical signal at a binding energy of about 530.0 eV, with a tail extending to the high binding energy. As an Ag layer deposited onto the $TiO_x$ film, the O 1s signal emerged from the surface was drastically depressed with a weak peak at about 531.1 eV. The O 1s signal measured from the surface of the 500 °C-annealed $TiO_x$/Ag structure was obviously enhanced with a peak of approximatively at 529.2 eV and a notable shoulder extending to the higher binding energy. Considering the reports on the oxidized states of the titanium and silver atoms, this signal emerging from the surface of the 500 °C-annealed $TiO_x$/Ag structure could be linked to a combination of the oxygen species of the Ag-O (~528.9 eV), Ti-O (~529.7 eV), terminal hydroxyl (~530.8 eV), surface carbonate-like admixtures (~531.7 eV), and adsorbed water [50,51,55–60]. According to the studies on the crystallinity and surface bond configurations of the $TiO_x$/Ag structure, it demonstrated that, although there was little chemical reaction between the $TiO_x$ film and Ag layer, the surface property combined with the oxidized state evolution from $Ti^{4+}$ to $Ti^{3+}$ in Ti-O bonds and the metallic ($Ag^0$)/oxidized Ag ($Ag^+$) states was measured from the 500 °C-annealed sample.

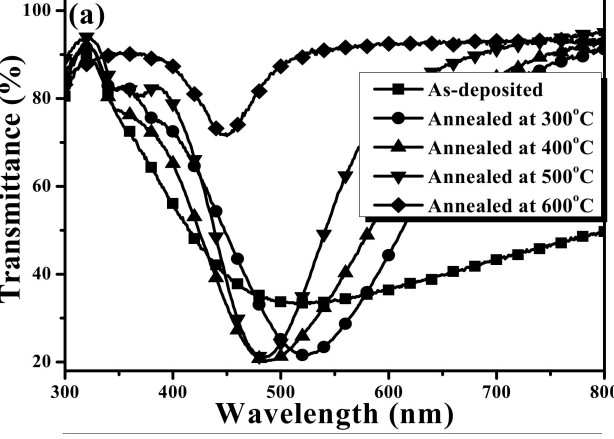

**Figure 3.** *Cont.*

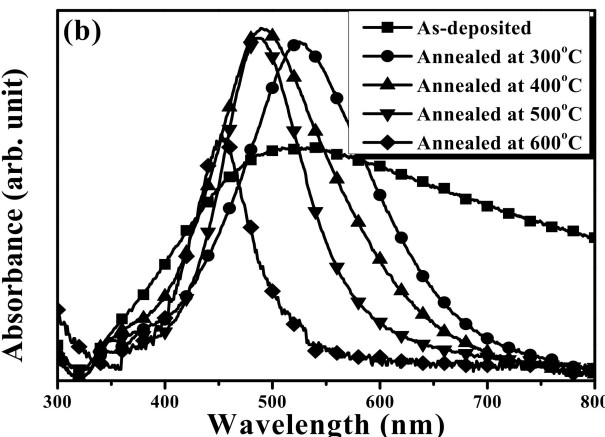

**Figure 3.** (**a**) Optical transmittance and (**b**) absorbance spectra of the as-deposited and annealed Ag films directly deposited onto the sapphire substrate.

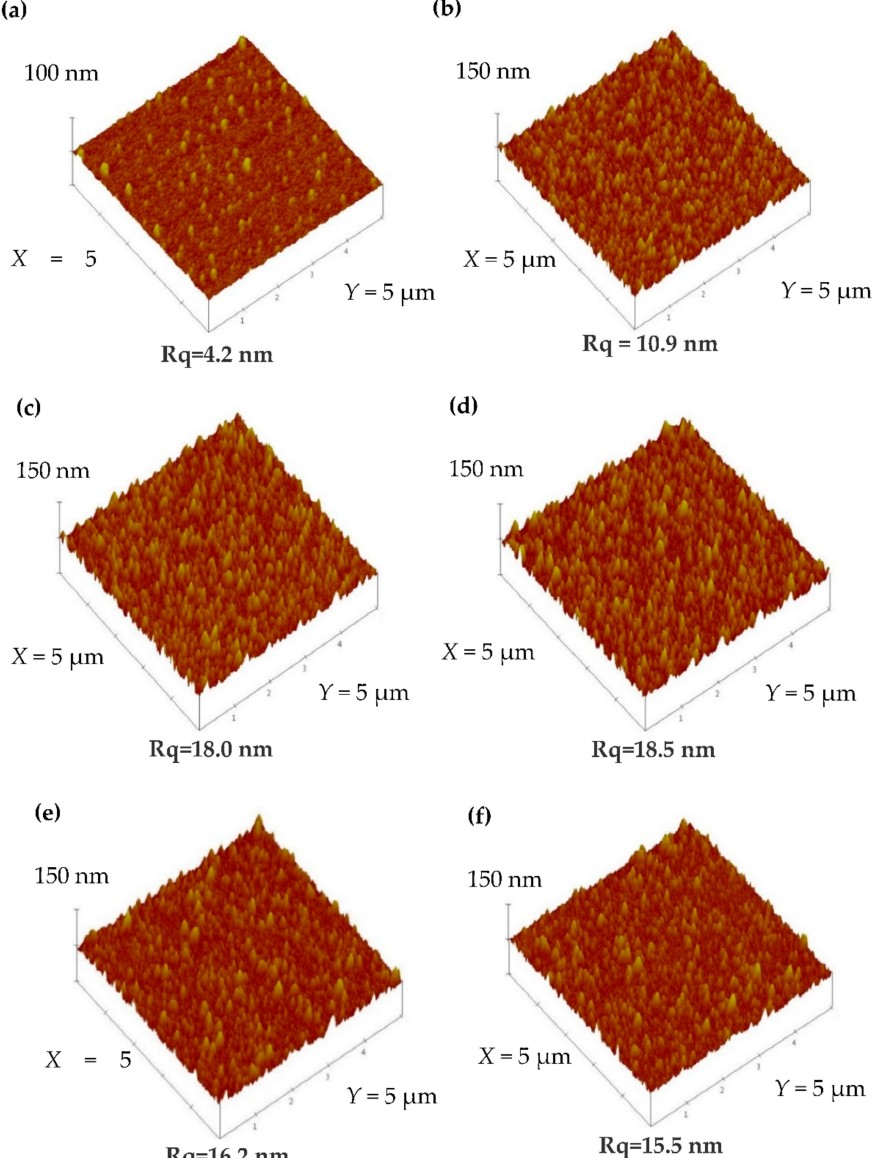

**Figure 4.** Surface roughness of (**a**) the TiO$_x$ film, (**b**) the as-deposited TiO$_x$/Ag structure, and the structures annealed at (**c**) 300, (**d**) 400, (**e**) 500, and (**f**) 600 °C under nitrogen ambient for 1 min, examined by AFM.

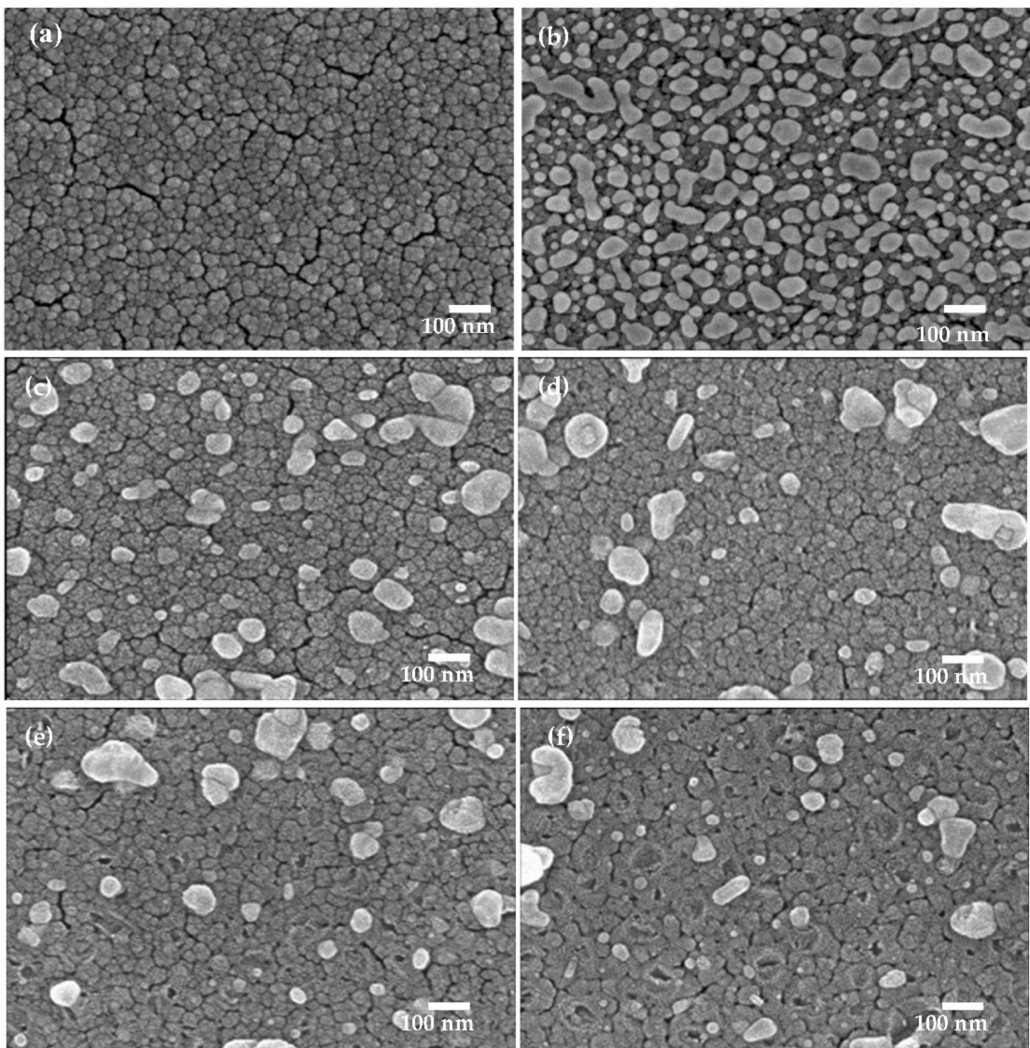

**Figure 5.** Surface morphologies of (**a**) the TiO$_x$ film, (**b**) the as-deposited TiO$_x$/Ag structure, and the structures annealed at (**c**) 300, (**d**) 400, (**e**) 500, and (**f**) 600 °C under nitrogen ambient for 1 min, conducted by FE-SEM observation.

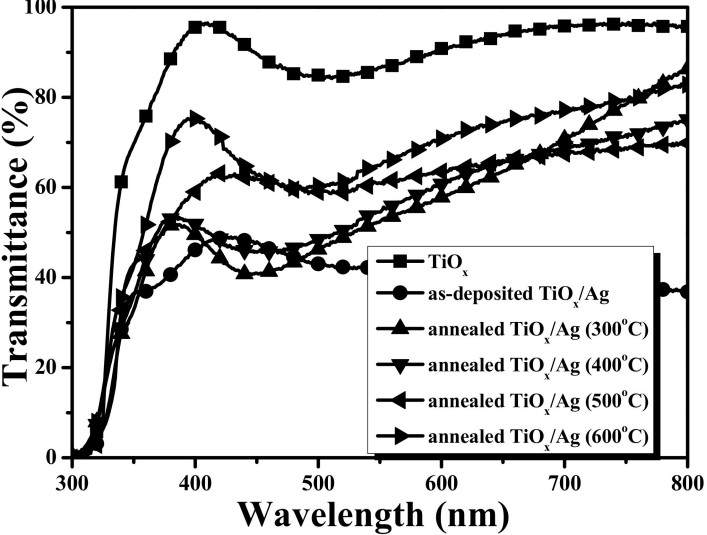

**Figure 6.** Optical transmittance of the as-deposited and annealed TiO$_x$/Ag structures (optical transmittance of the TiO$_x$ film also is shown as a comparison).

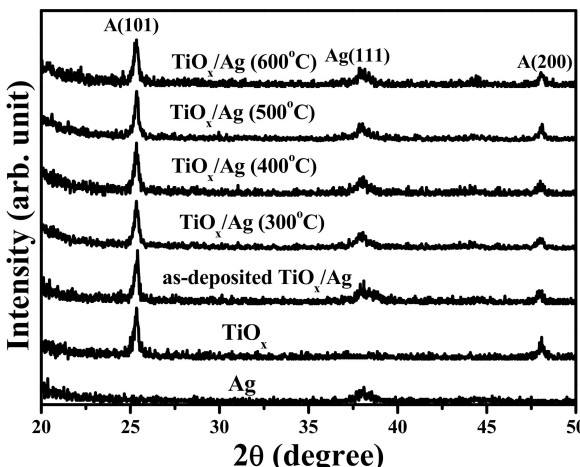

**Figure 7.** XRD spectra of the as-deposited and annealed $TiO_x$/Ag structures as well as the $TiO_x$ and Ag films.

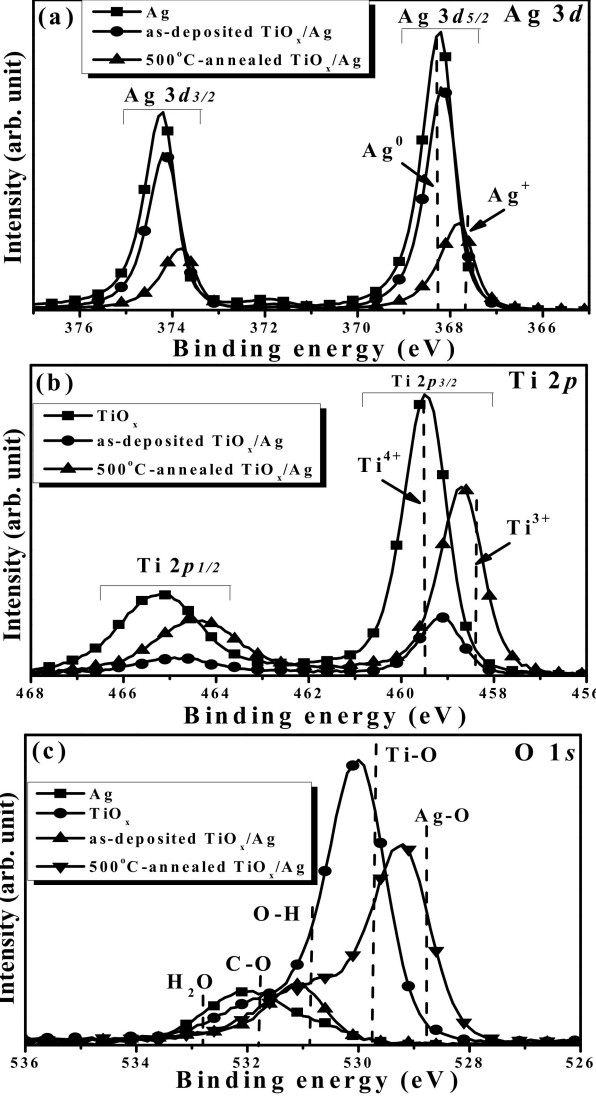

**Figure 8.** Binding energy spectra of the (**a**) Ag 3d, (**b**) Ti 2p, and (**c**) O 1s core levels measured from the surface of the as-deposited and 500 °C-annealed $TiO_x$/Ag structures (the individual Ag and $TiO_x$ films are also given as comparisons).

The concentration evolution of the MB solution decomposed by using the TiO$_x$ film, thin Ag layer, and the as-deposited and annealed TiO$_x$/Ag structures under the UV light irradiation are illustrated in Figure 9a (the change in the concentration of the MB solution directly irradiated by the UV light is also provided as a comparison). The degree of the photocatalytic activity for these samples, determined in terms of the rate constant, is summarized in Table 1. The rate constant, $k$, is derived by fitting the curves in Figure 9a, using the following relationship [61]:

$$\ln(C/C_0) = kt \tag{1}$$

where $C$ and $C_0$ are the concentrations of the MB solution under UV light irradiation time of $t = 0$ and $t$, respectively. As can be seen in this figure, the degradation of the MB solution induced by the UV light irradiation was significantly enhanced by introducing a TiO$_x$ film, while the Ag layer was almost useless in strengthening the decomposition of the MB solution under UV light irradiation. The corresponding rate constant derived from the curve of the Ag layer (~0.0029 min$^{-1}$) was only a little higher than that of the curve derived from the MB solution (~0.0027 min$^{-1}$). In contrast to the Ag layer, a nearly quadruple increase in the rate constant (~0.0102 min$^{-1}$) was obtained from the MB solution decomposed by the TiO$_x$ film under UV light irradiation. Interestingly, although the Ag layer individually was inactive in decomposing the MB solution, the as-deposited TiO$_x$/Ag structure irradiated by the UV light took effect and corresponded to a rate constant of about 0.0110 min$^{-1}$. As referred to in Figure 5b, the areas of the TiO$_x$ film uncovered by the Ag layer might be responsible for the activation regions to decomposed the MB solution. Moreover, the appearance of the Ti$^{3+}$ state on the structural surface induced by the Ag ions which functioned to suppress the recombination of the photogenerated ehps resulted in a rate constant being higher than that of the individual TiO$_x$ film. For the same reason, the extensive reaction area of the TiO$_x$ surface that was abundant in the Ti$^{3+}$ chemical state for the 500 °C-annealed TiO$_x$/Ag structure, as demonstrated in Figures 5e and 8b, resulted in the significant enhancement on the photodecomposition to the MB solution at a rate constant of about 0.0141 min$^{-1}$. The degree of the rate constant for the TiO$_x$/Ag nanoparticles system evaluated from decolorizing the MB solution was comparable to that of the photocatalytic materials under sunlight or UV light irradiation, as quoted from the reports [62–66]. The photo-excited current responses as the transient of the UV light irradiation for the TiO$_x$ film and 500 °C-annealed TiO$_x$/Ag structure are given in Figure 9b. According to the cycles of the on-off current transient, the photocatalysis of the samples under UV light irradiation was reproducible and reliable. Under the same irradiation power, the degree of the photo-induced current from the TiO$_x$/Ag structure was much higher than that of the TiO$_x$ film, indicating that the TiO$_x$/Ag structure was beneficial for the separation of the photo-generated ehps. Moreover, the longer transient time determined from the highest photo-excited current to the dark current was obtained from the TiO$_x$/Ag structure (8 min) as compared to that of the TiO$_x$ film (4 min). This also showed evidence of the recombination rate of the photo-generated ehps could be effectively alleviated in the TiO$_x$/Ag structure. The mechanism responsible for the separation of the photo-generated ehps is ascribed to the difference of the electron affinity between the TiO$_x$ and Ag interface which causes the photo-generated electrons flow to the Ag surface while the photo-generated holes stay at the TiO$_x$ side [33]. In addition, the presence of the Ag nanoparticles that accept oxygen atoms from the TiO$_x$ matrix to result in the reduction of the oxidized state of Ti$^{4+}$ to Ti$^{3+}$ is also facilitated for capturing the photo-generated hole on the TiO$_x$ surface [46]. Those photo-generated ehps react with the adsorbed O$_2$ and H$_2$O and subsequently produce free radicals of ·OH and ·O$_2$$^-$, thereby decomposing the MB solution into nontoxic products [62]. Figure 10a–d show the photographs of the *E. coli* bacterial colonies on the nutrient agar after a 24 h incubation period using the UV light-activated as-deposited and annealed TiO$_x$/Ag structure as well as using the TiO$_x$ film and thin Ag layer, respectively. The number of the bacterial colonies counted from the plate, as shown in Figure 10a,b, was about $(230 \pm 11) \times 10^3$ CFU/mL and $(215 \pm 6) \times 10^3$ CFU/mL,

respectively. As the *E. coli* was sterilized using the as-deposited $TiO_x$/Ag structure, the number of the bacteria grown on the plate significant decreased to $(133 \pm 8) \times 10^3$ CFU/mL. Eventually, the bacteria were almost absent on the plate as they were sterilized by using the 500 °C-annealed $TiO_x$/Ag structure. The reduction percentage, *R*, of the bacteria growth on the plate was calculated using the following equation [67]:

$$R = [(A - B)/A] \times 100\% \qquad (2)$$

where *A* is the number of the bacterial colonies counted from the *E. coli* directly sterilized by the UV-lamp irradiation (~$382 \pm 16 \times 10^3$ CFU/mL), and *B* is the number of the bacterial colonies counted from the plate treated by the samples. The reduction percentages of these plates shown in Figure 10 are given in Table 1. The reduction percentages of the *E. coli* using the Ag and $TiO_x$ films were about 44% and 40 %, respectively. The sterilized activities conducted from the Ag layer was ascribed to the release of the $Ag^+$ ion could bind the proteins of the bacteria to cause the increase of the cellular oxidative stress, thereby resulting in the protein deactivation [68]. On the other hand, the photo-generated ehps on the $TiO_x$ surface could produce active radicals of ·OH and ·$O_2^-$ to interact with cell walls and rupture bacterial cell membranes [69]. Accordingly, the $TiO_x$ film incorporating a thin Ag layer resulted in the antibacterial efficiency against *E. coli*, with a higher reduction percentage of 65%. Moreover, the achievement of the $TiO_x$ film surface decorated with the Ag nanoparticles which was abundant in the $Ag^+$ and oxidant radical such as ·OH and ·$O_2^-$ on the surface as it was annealed at a temperature of 500 °C for 1 min under nitrogen ambient could completely inhibit the growth of the *E. coli* bacteria.

**Table 1.** Rate constant (*k*) and reduction percentage (*R*) of the Ag, $TiO_x$, as-deposited $TiO_x$/Ag, and 500 °C-annealed $TiO_x$/Ag samples.

| | Ag | $TiO_x$ | As-Deposited $TiO_x$/Ag | 500 °C-Annealed $TiO_x$/Ag |
|---|---|---|---|---|
| $k$ (min$^{-1}$) | 0.0029 | 0.0102 | 0.0110 | 0.0141 |
| $R$ (%) | 44 | 40 | 65 | 100 |

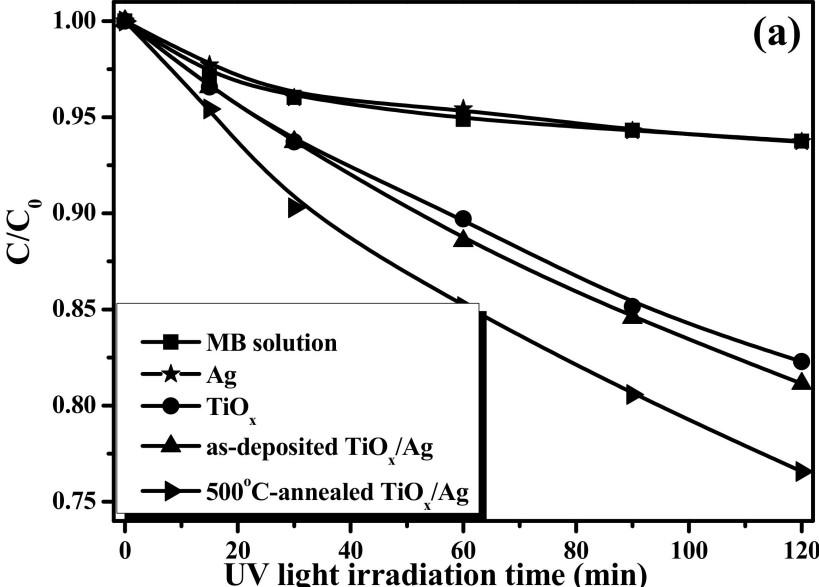

**Figure 9.** *Cont.*

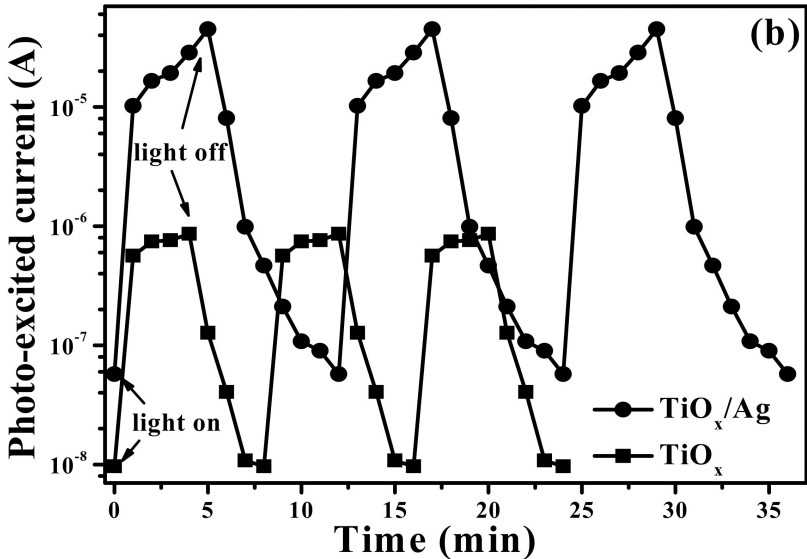

**Figure 9.** (**a**) Concentration evolution of the MB solution decomposed by using the TiO$_x$ film, thin Ag layer, and the as-deposited and annealed TiO$_x$/Ag structures under the UV light irradiation (the change in the concentration of the MB solution directly irradiated by the UV light is also provided as a comparison); (**b**) the photo-excited current responses as the transient of the UV light irradiation for the TiO$_x$ film and 500 °C–annealed TiO$_x$/Ag structure.

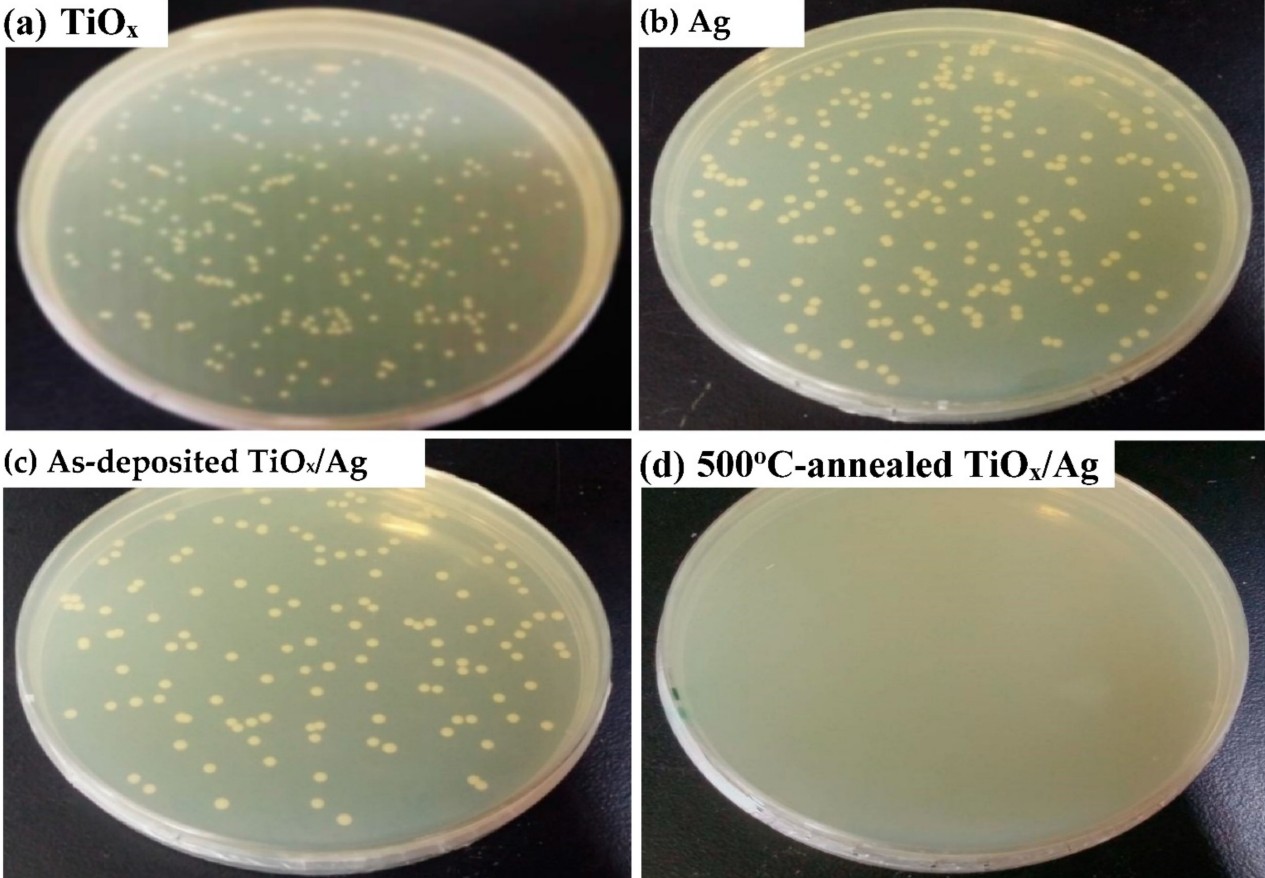

**Figure 10.** Photographs of the *E. coli* bacterial colonies on the nutrient agar after a 24 h incubation period that are sterilized using the (**a**) TiO$_x$, (**b**) Ag, (**c**) as-deposited TiO$_x$/Ag and (**d**) annealed TiO$_x$/Ag samples under the UV-lamp irradiation for 1 h.

## 4. Conclusions

The surface property of the $TiO_x/Ag$ structure activated by a UV-lamp irradiation was investigated. A thin Ag layer deposited onto the anatase-$TiO_x$ film exhibited an uneven surface morphology with separated Ag nanoparticles and wedge-liked islands. This $TiO_x/Ag$ structure annealed under nitrogen ambient caused the agglomeration and diffusion of the Ag atoms, resulting in extensive $TiO_x$ film being exposed on the surface. The chemical bond configurations associated with the metallic $Ag^0$ to oxidized $Ag^+$ and the reduced oxidized states of $Ti^{4+}$ to $Ti^{3+}$, which were beneficial to separate the photogenerated ehps and suppress the recombination rate, were observed from the composite surface of the Ag nanoparticles and $TiO_x$ layer. The decomposition efficiency of the MB solution under UV-lamp irradiation was apparently improved incorporating with the $TiO_x/Ag$ structure annealed at a temperature of 500 °C under nitrogen ambient for 1 min. The derived rate constant (~0.0141 $min^{-1}$) was about 40% higher than that of the rate constant as incorporated with only the $TiO_x$ film (~0.0102 $min^{-1}$). This $TiO_x$ film decorated by the Ag nanoparticles which produced hydroxyl radicals and reactive oxygen species on the $TiO_x$ surface under UV light irradiation as well as the release of the $Ag^+$ ions from the nanoparticles resulted in the enhanced inhibition of the growth of the *E. coli*. The reduction percentage against the *E. coli* using the as-deposited $TiO_x/Ag$ structure was about 65%, whereas a perfect antibacterial efficiency (the reduction percentage reaches 100%) was achieved by using the 500 °C-annealed sample. Such a surface property possessed excellent photo-activated properties to decompose pollutants and inhibit the growth of the microbial organisms, providing promising high-value products for the purification demands related to environmental pollution and sanitation.

**Author Contributions:** Conceptualization, D.-S.L.; Resource, D.-S.L.; Methodology, K.-S.Y.; Formal analysis, P.-J.S.; Investigation, P.-J.S., J.-K.Z. and C.-H.H.; Writing-original draft, J.-K.Z. and C.-H.H.; Writing-review & editing, D.-S.L. All authors have read and agreed to the published version of the manuscript.

**Funding:** This research was funded by Ministry of Science and Technology grant MOST 109-2622-E-150-023 and also was fund by ITRI and MIRDC.

**Institutional Review Board Statement:** Not applicable.

**Informed Consent Statement:** Not applicable.

**Data Availability Statement:** Not applicable.

**Conflicts of Interest:** The authors declare that there are no conflict of interest regarding the publication of this paper.

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
