# Peer review of "Investigations into the Photocatalytic and Antibacterial Activity of the Nitrogen-Annealed Titanium Oxide/Silver Structure"

_coatings, doi:10.3390/coatings12111671_

Round 1

Reviewer 1 Report

Manuscript by Zhang  et. al., describes in this study, a thin silver (Ag) layer was evaporated onto the anatase-titanium oxide (TiOx) film. This structure was then annealed at various temperatures under nitrogen ambient to realize the formation of the Ag nanoparticles. The photocatalytic activities of these structures were determined from the rate constant while they were applied to decompose the methylene blue solu-tion in the presence of the UV light irradiation. An increase of about 40% in the rate constant compared to the individual TiOx film was obtained from the 500 oC-annealed TiOx/Ag structure. The enhancement on the photocatalyticity was ascribed to the presence of the Ag nanoparticle being facilitated for the transformation of the Ti4+ into the Ti3+ state in the TiOx film, thereby alleviating the recombination of the photogenerated electron-hole-pairs. Moreover, as a consequence of the surface composed of Ti3+ and Ag+ chemical states, this TiOx film incorporating with the Ag nanoparticles also showed a perfect antibacterial efficiency against Escherichia coli. After reading the manuscript I feel that the topic is interesting but the execution and presentation of work needs to highlight the novelty the author should carefully address following comments:

1.     Rewrite abstract it is too short present some prominent results in it.

2.     In introduction section author needs to discuss more about antimicrobial and why photocatalysis is better than the other methods for dye removal such as ion exchange, adsorption etc and also discus different type of photocatalyst flow following works Microchemical Journal 149 (2019) 103966; Chemical Engineering Journal 251 (2014), 413-421.

3.     Please add in detail about photocatalysis part into experimental section not as theoretical.

4.     Author has not supported characterization aspect with proper reference specially xrd please use more relevant works.

5.     Quality of all figures need to be improved.

6.     To demonstrate stability of material please add some studies or provide SEM of material after photocatalysis.

7.     The photocatalysis mechanism need to be elaborated more please refer related works Environmental Pollution 308 (2022), 119597.

8.     Clearly state how doping affects photodegradation rate.

9.     If possible provide elemental mapping or edx of materials.

10.  Present only prominent results in conclusion section.

11.  Compare your photodegradation results of MB with other reported works in tabular form see Journal of Industrial and Engineering Chemistry 21 (2015), 957-964; Journal of Industrial and Engineering Chemistry 20 (2015), 3596-3603.

Author Response

The point-to-point responses for the reviewer's comments are list in separated sheets.

Reviewer 2 Report

The following issues must be addressed:

1.     Introduction part should be improved by inserting more representative references (i.e. DOI: 10.1016/j.physe.2020.114236; DOI: 10.1016/j.apsusc.2011.12.110; DOI: 0.1016/j.mssp.2021.105923) and outlining what is new and innovative in this work compared with other similar studies;

2.     Provide the purity of each substance involved in this study;

3.     Photocatalyst dosage must be provided;

4.     Light irradiance (mW/cm2) in the sample proximity must be provided;

5.     The authors should explain why MB was considered for photocatalytic study.  There are already many studies about MB photodegradation.

6.     SBET must be provided for each sample.

7.     Please explain in more details why the annealing process improve the photocatalytic properties.

Author Response

(The authors gave the same response as above.)

Reviewer 3 Report

The reviewer found the idea of the submitted manuscript, the title ‘Investigations into the Photocatalytic and Antibacterial Activity of the Nitrogen-Annealed Titanium Oxide/Silver Structure’ interesting and can be published in “Coatings.” However, it is needed to be revised according to the following comments with a minor revision before the publication.

1.      Authors must adopt the same font style and spacing before their final submission. Sometimes they used Italian in normal writing. Concise and revise the conclusion part. Figures usually look more attractive and self-explanatory in colored form. I would recommend the authors as well. Furthermore, in the text, It needs to use the same wording, either Figure/Figures or Fig./Figs, throughout the whole manuscript. The Legend frame position does not look nice. Please reconsider the figures with an appropriate position of legends that do not affect the graph information.  

2.      Author mentioned that in the experiment, they used the UV lamp of wavelength 365 nm, while UV has in the usual range of 100 to 400 nm; why this specific wavelength was employed?

3.      In Fig. 7, there is a little unclear peak around 44.5 (2-theta scan).

4.      In Binding energy spectra, peak shifting is a very common and obvious process in forming oxides/compounds, which can even be seen here. However, the readers might be curious about the explanation/reasoning concisely. Please include the requested information briefly in the text. It would be nice if the authors could investigate the O1-s peak into three well-known asymmetric O-1s peaks in depth. Otherwise, they can add at least the following references as recently published reports in the manuscript to better understand the readers.

(a)    Ultraviolet Photodetection Based on High-Performance Co-Plus-Ni Doped ZnO Nanorods Grown by Hydrothermal Method on Transparent Plastic Substrate 

(b)   High-performance flexible ultraviolet photodetectors with Ni/Cu-codoped ZnO nanorods grown on PET substrates

5.      Do not overlap the Legend frame position with the graphical information. 

Author Response

(The authors gave the same response as above.)

Reviewer 4 Report

The paper presents original and new ideas and can be accepted for publication after major revision. The following issues should be addressed. 

Please widen the introduction's discussion about the photocatalytic and antibacterial activity of similar structures. In my opinion, too few papers on this topic were mentioned. 

Information in captions to Fig. 1-2 and 4-5 should be pointed out in detail. If I understand rightly Fig.1 and 4 depict the topography of the coatings and  Fig. 2 and 5 the phase analysis. Why different scales are presented for these images? What is depicted in Fig. 1 and 4? Are the spheroids presented here silver nanostructures? If yes, why the shape of the silver nanostructures deposited on the different surfaces is different? Maybe it will be valuable to add SEM images. 

Please add information about the mechanism of the antibacterial action presented here system.

The captions to the figures are partially cut off.

I suggest to cite relevant papers where similar systems were presented:

https://doi.org/10.1016/j.cej.2022.137048

Author Response

(The authors gave the same response as above.)

Round 2

Reviewer 2 Report

The manuscript can be published in present form.

Reviewer 4 Report

The authors have answered all of my issues and the paper can be accepted in its present form.